# Unsupervised motion segmentation in one go: Smooth long-term model over a video

## Abstract

Human beings have the ability to continuously analyze a video and immediately extract the main motion components. Motion segmentation methods often proceed frame by frame. We want to go beyond this classical paradigm, and perform the motion segmentation over a video sequence in one go. It will be a prominent added value for downstream computer vision tasks, and could provide a pretext criterion for unsupervised video representation learning. In this perspective, we propose a novel long-term spatio-temporal model operating in a totally unsupervised way. It takes as input the volume of consecutive optical flow (OF) fields, and delivers a volume of segments of coherent motion over the video. More specifically, we have designed a transformer-based network, where we leverage a mathematically well-founded framework, the Evidence Lower Bound (ELBO), to infer the loss function. The loss function combines a flow reconstruction term involving spatio-temporal parametric motion models combining, in a novel way, polynomial (quadratic) motion models for the $(x, y)$-spatial dimensions and B-splines for the time dimension of the video sequence, and a regularization term enforcing temporal consistency on the masks. We report experiments on four VOS benchmarks with convincing quantitative results. We also highlight through visual results the key contributions on temporal consistency brought by our method.

## 1 Introduction

When dealing with videos, motion segmentation is one of the key issues. Human beings have the ability to continuously analyze a video and immediately extract the main motion components. Computer vision methods usually proceed frame by frame. We want to go beyond this classical paradigm and segment the different motion entities in a video sequence in one go. We thus thoroughly investigate the temporal dimension of the motion segmentation problem. Since the optical flow (OF) yields all the information on the movement between two images of the video sequence, it is natural to base motion segmentation on optical flow. Motion segmentation is a computer vision task in its own, but it is also useful for diverse downstream tasks as independent moving object detection, object tracking, or action recognition, to name a few. It is also leveraged in video object segmentation (VOS), while most often coupled with appearance.

The optical flow field at time $t$ enables to get the segmentation at frame $t$ of the video. However, taking a large time window, and even the whole video sequence, is beneficial, since motion entities are generally consistent throughout a video sequence (or at least within every video shot for long videos). Indeed, temporal coherence is inherent to motion. Therefore, it is essential that motion segmentation takes advantage of it, especially in a long-term perspective.

In this paper, we propose an original holistic method for multiple motion segmentation from optical flow over a video sequence. To the best of our knowledge, our optical flow segmentation (OFS) method is the first fully unsupervised network to involve long-term temporal consistency, and to segment multiple motions in a video sequence in one go. The main contributions of our work are as follows. Our network takes as input a volume of consecutive optical flows, and delivers consistent motion segmentation maps throughout the video sequence. It involves a transformer module allowing for long-term interactons. It is trained in a completely unsupervised manner, without any manual annotation or ground truth data of any kind. The loss function is inferred by leveraging the Evidence Lower Bound (ELBO) framework, and comprises a flow reconstruction term with original

spatio-temporal parametric motion models and an additional term enforcing temporal consistency on the segmentation masks. We model with B-splines the long-term temporal evolution of the motion model parameters. Our method also involves a latent representation of the segment motion augmented with positional embedding.

The rest of the paper is organized as follows. Section 2 describes related work regarding motion segmentation. Section 3 presents our unsupervised network for multiple motion segmentation in one go, embedding long-term temporal consistency. In Section 4, we provide implementation details. Section 5 reports results on four VOS benchmarks with a comparison to state-of-the-art unsupervised motion segmentation methods. Finally, Section 6 contains concluding remarks.

## 2 RELATED WORK

Motion segmentation aims to break down each frame of a video sequence into components (or segments) of coherent motion. Usually, each motion segment is identified by a motion model, which can be hand-crafted such as affine or quadratic polynomial models or sometimes learned. Motion segmentation has been investigated for decades (19; 23; 38). However, we will focus on recent methods in this section. Since accurate and efficient methods for estimating optical flow are now available, motion segmentation methods can leverage optical flow as a reliable input. The advent of deep learning methods in computer vision, and more recently the use of transformer-based networks with attention mechanisms (11), has also encompassed motion segmentation.

In (35), a transformer module, more specifically, the slot attention mechanism introduced in (18), is leveraged to perform motion segmentation from the optical flow. As a matter of fact, it addresses a binary segmentation problem, foreground moving object *vs* background. The loss function is composed of two terms, a flow reconstruction term and an entropy term to make masks as binary as possible. Another approach is adopted in (34) that is able to cope with multiple motion segmentation. Nonlinear subspace filters are learned from stacked deep multi-layer perceptrons. Then, motion segmentation is inferred at inference by applying $K$-means to the output embeddings. In (21), the Expectation-Maximization (EM) framework is leveraged to define the loss function and the training procedure for an unsupervised frame-by-frame motion segmentation, where 12-parameter quadratic motion models are involved. The same authors also proposed an extension of that method in (22) to better handle the temporal dimension of the problem, by considering triplets of flows as input. In (36), the authors developed an adversarial method whose aim is to generate a mask hiding the input optical flow, where an inpainter network attempts to recover the flow within the mask. The rationale is that no correct reconstruction of the inside flow can be achieved from the outside flow, if the hidden area exhibits an independent motion and then constitutes a motion segment.

The temporal dimension of motion segmentation has been considered in various ways (30). Regarding deep learning approaches, motion segmentation at time $t$ is improved at training time in (35), by taking, as input, flows between $t$ and several time instants before and after $t$. The authors of (8) consider several consecutive RGB frames as input of their self-supervised method. Optical flow is only computed at training time, and the loss function also comprises a temporal consistency term. However, the latter is not applied to two consecutive segmentation masks, but for pairings between the same frame $t$ and another (more or less distant) one. In (9), spatio-temporal transformers were designed for video object segmentation involving temporal feature propagation.

VOS usually requires motion segmentation, often coupled with appearance (5; 10; 37). It was addressed with supervised or semi-supervised methods as in (7; 9; 12), but also with unsupervised methods (36; 35; 21). VOS is concerned with the segmentation of primary objects, i.e., object moving in the foreground of a scene and possibly tracked by the camera (39). Thus, VOS generally involves binary ground truth segregating primary moving object against background. This is for instance the case for the DAVIS2016 benchmark (26). Recent works have revisited the way of coupling appearance and motion for VOS. The AMD method (17) includes two pathways, the appearance one and the motion one. If it does not use optical flow as input and brings out the objectness concept, it nevertheless relies on a coarse motion representation. The RCF method (16) involves learnable motion models, and is structured in two stages, a motion-supervised object discovery stage, and then, a refinement stage with residual motion prediction and high-level appearance supervision. However, the method cannot distinguish objects undergoing different motions. In (6), the prediction of probable motion patterns is used at the training stage as a cue to learn objectness from videos. Divided

attention is promoted in (14). The resulting DivA method is based on the same principle as in (36) that motion segments are mutually uninformative. However, it is not limited to binary segmentation. It can segment a variable number of moving objects from optical flow, by leveraging a slot attention mechanism guided by the image content through a cross-modal conditional slot decoder.

Our fully unsupervised approach differs from these previous works in several respects. We rely only on optical flow and take a volume of OF fields as input, providing a volume of consistent segmentation maps. We introduce B-splines to define parametric motion models able to correctly handle motion evolution over time, and we express long-term temporal consistency. We infer the loss function from the Evidence Lower Bound (ELBO) framework. In (22), the authors also deal with temporal consistency. However, they only took a triplet of optical flow fields as input, they did not resort to transformers, and the temporal part of the motion model was just a linear (first-order polynomial) model in time. In addition, their temporal linkage of the consecutive motion segmentations over a video sequence was an *ad hoc* post-processing. In contrast, we define a fully integrated long-term model that is end-to-end trained in a unsupervisedly way. We will call our method LT-MS method, for long-term motion segmentation, and in contrast, we will refer to the method (22) as the ST-MS method, for short-term motion segmentation.

## 3 LONG-TERM MOTION SEGMENTATION METHOD

We have designed a transformer-based network for multiple motion segmentation from optical flow. It is inspired from the MaskFormer architecture (4), but it only comprises one head corresponding to the mask prediction, as described in Fig.1. The network takes as input a volume, of flexible temporal length, comprising several consecutive optical flow fields. Temporal consistency is expressed in two main ways at the training stage. Firstly, we associate a space-time motion model with each segment to characterize its motion along with the evolution of the latter over time. Secondly, the loss function comprises an additional term enforcing consistent labeling of the motion segments over the volume.

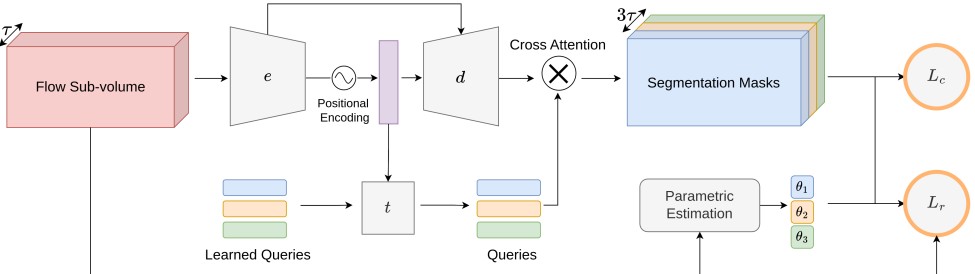

Figure 1: Overall architecture of our multiple motion segmentation method ensuring temporal consistency with the loss term $\mathcal{L}_c$ and the B-spline space-time motion models $\theta_k$ (for $k = 1, .., K$). It takes as input a volume of $T$ flow fields. It comprises a 3D U-net ($e$ and $d$ boxes) and a transformer decoder ($t$ box). It also involves positional encoding. A cross-attention product yields the $K$ segmentation masks corresponding to the input volume For the sake of clarity, the block diagram is represented for three motion segments ($K = 3$). $\mathcal{L}_r$ is the flow-reconstruction loss term.

### 3.1 SPATIO-TEMPORAL PARAMETRIC MOTION MODEL

The set of $T$ consecutive flows will be designated as a space-time volume (or volume, to make it short). The volume could even be the whole video sequence. Our space-time motion model is estimated through B-spline functions (32). We assign a spatio-temporal parametric motion model $\tilde{f}_{\theta_k}$ to each motion segment $k, k = 1, .., K$. $\theta_k$ specifies the motion model $\tilde{f}$ for segment $k$. The motion model involves $J$ parameters and each parameter $\theta_{k_j}, j = 1, .., J$, of the model results from a B-spline function of order $n$ in the variable $t$ over the space-time volume. In practice, we take $n = 3$. The number $L$ of control points is given by $L = 2 + \lfloor \frac{T-2}{\nu} \rfloor$, where $\nu$ allows us to set the temporal frequency of control points. We put a control point at both ends of the volume (at $t = 1$ and $t = T$), and the other control points are equidistantly placed between the two. Most often, the control points are not located at time points of the video sequence.

The space-time spline-based motion model is illustrated in Fig.2. Just for this illustration, the motion models are computed within the two segments, foreground moving object and background, provided by the ground truth. The estimated motion model for the foreground moving object is able to capture the periodic nature of the swing motion as demonstrated by the plots of the computed motion model parameters. Also, the motion model computed in the background segment perfectly fits the camera motion. Articulated motion (woman's legs) would require multiple-motion segmentation.

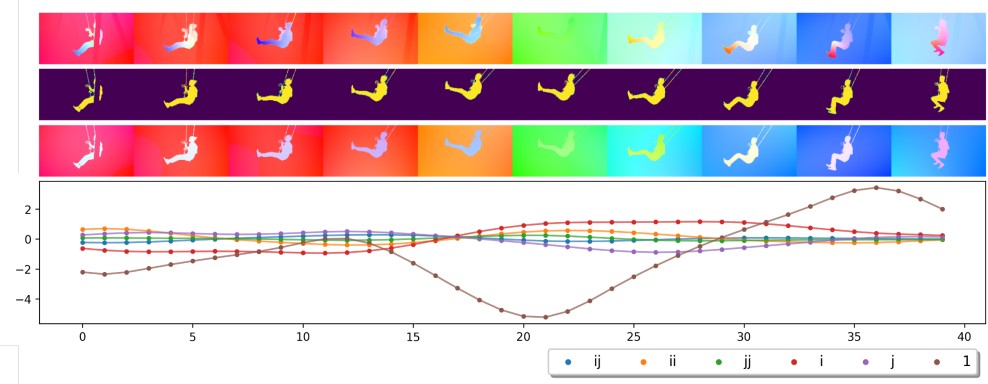

Figure 2: Illustration of the spatiotemporal spline-based motion model. Top row: input flows displayed with the HSV code for the *swing* video of DAVIS2016 dataset, binary segmentation ground truth, flows generated by the estimated spline-based motion models for the two segments. Bottom row: plot of the temporal evolution of the six estimated model parameters corresponding to the flow $u$-coordinate for the foreground moving object.

Any parametric motion model could be considered. We use the 12-parameter quadratic motion model to be able to account for continuously varying depth surface of the objects in the scene, especially for the whole background, and for complex object or camera motions. In contrast, the affine and the 8-parameter quadratic motion models assume a planar object surface. Indeed, the latter exactly corresponds to the projection in the image of the 3D rigid motion of a planar surface. It is equivalent for velocity fields to the homography transform. However, in presence of severe depth effects (strong depth discontinuities) and camera motion, the static scene cannot be represented by a single motion model due to motion parallax produced by static objects located in the foreground. Regarding first the spatial dimension of the motion model, the 2D flow vector yielded by the full quadratic motion model at point $(x, y)$ writes:

$$\tilde{f}_\theta(x, y) = (\theta_1 + \theta_2 x + \theta_3 y + \theta_7 x^2 + \theta_8 xy + \theta_9 y^2, \theta_4 + \theta_5 x + \theta_6 y + \theta_{10} x^2 + \theta_{11} xy + \theta_{12} y^2)^T. \quad (1)$$

To correctly handle complex temporal evolution, we resort to the spline approximation as aforementioned. By denoting now $S_n(\theta_k)$ the subscript of $\tilde{f}$, we emphasize that the motion model parameters for each segment $k$ are estimated through the B-spline functions. More specifically, we have:

$$\tilde{f}_{S_n(\theta_k)}(i, t) = \sum_{l=1}^{L} \tilde{f}_{\theta_{k,l}}(i, t) B_{n,l}(t), \quad (2)$$

where $B_{n,l}$ is the $l^{th}$ B-spline and $\theta_{k,l}$ corresponds to the $l^{th}$ control point of the spline function.

## 3.2 Loss function

We consider a volume of optical flow fields $f \in \mathbb{R}^{2 \times T \times W \times H}$, the spatial grid $\Omega \in \mathbb{R}^{W \times H}$ and $T$ temporal steps. We denote $f(i, t) \in \mathbb{R}^2$ the flow associated to site $i \in \Omega$ at time $t$. We assume that we can decompose the flow as a set of $K$ segments, each one exhibiting a coherent motion. Flow vectors within a given segment $k$ are represented by a smooth parametric motion model parametrized by $\vartheta_k$. Variable $z_{i,t}$ conveys the motion segmentation, $z_{i,t}^k = 1$ if site $(i, t)$ belongs to segment $k$. $z$ and $\vartheta = \{\vartheta_k, k = 1 \cdots K\}$ are latent variables, and $f$ is the observed data. Following (13), we introduce an approximate distribution over segmentation and motion model parameters:

$$q(z, \vartheta | f) = q(z | f) q(\vartheta) = (\prod_{t=1}^{T} \prod_{i \in \Omega} \prod_{k=1}^{K} q(z_{i,t}^k | f))(\prod_{k}^{K} q(\vartheta_k)), \quad (3)$$

where $q(\vartheta_k) \triangleq \delta(\theta_k)$ with $\delta$ the dirac distribution and $q(z_{i,t}^k|f) = g_\phi(f)_{i,t}^k$. $g_\phi$ is our network model taking as input the optical flow volume and returning a probabilistic segmentation volume.

We can also write the data-likelihood over a dataset $\mathcal{D}$ of optical flow volumes $f$ as:

$$
\log(\mathcal{D}) = \sum_{f \in D} \log p(f) = \sum_{f \in D} \left( \mathbb{E}_{q(z,\vartheta|f)}\left[\log \frac{p(f,z,\vartheta)}{q(z,\vartheta|f)}\right] + KL(q(z,\vartheta|f)||p(z,\vartheta|f))] \right)
$$
$$
\geq \sum_{f \in D} \left( \mathbb{E}_{q(z,\vartheta|f)}[\log p(f|z,\vartheta)] - KL(q(z,\vartheta|f)||p(z,\vartheta))) \right), \quad (4)
$$

where we obtain the Evidence Lower Bound (ELBO). Following the assumption stated above, we can write our likelihood as:

$$
p(f|z,\vartheta) = \prod_{t=1}^{T}\prod_{k=1}^{K} p(f(t)|\vartheta_k, z) \propto \prod_{t=1}^{T}\prod_{k=1}^{K}\prod_{i\in\Omega} \exp(-||f(i,t) - \tilde{f}_{S_n(\vartheta_k)}(i,t)||_1/\xi_{f,t})^{z_{i,t}^k}, \quad (5)
$$

where $\tilde{f}_{S_n(\vartheta_k)}(i,t)$ is the flow vector given by the parametric motion model of parameters $\vartheta_k$ at point $(i,t)$ as defined in Section 3.1 through the spline approximation and $\xi_{f,t} = \sum_i ||f(i,t)||_1$ is a normalizing factor. From this likelihood, we can write the left term of the ELBO as:

$$
\mathcal{L}_r = \mathbb{E}_{q(z,\vartheta|f)}[\log p(f|z,\vartheta)] = \sum_{t=1}^{T}\sum_{k} \mathbb{E}_q[\log p(f(t)|\vartheta_k, z)]
$$
$$
= -\sum_{t=1}^{T}\sum_{i\in\Omega}\sum_{k} g_\phi(f)_{i,t}^k ||f(i,t) - \tilde{f}_{S_n(\theta_k)}(i,t)||_1/\xi_{f,t} \quad (6)
$$

The term $KL(q(z,\vartheta|f)||p(z,\vartheta)))$ allows us to define a prior over the segmentation. We want to enforce the property that the segmentation labels are temporally consistent, i.e., similar between neighbouring frames. We approximate this term using the temporal consistency loss defined in (22), as it showed good experimental results and was efficient to compute at training time:

$$
\mathcal{L}_c = \frac{1}{2K|\mathcal{I}|} \sum_{i\in\Omega}\sum_{t=2}^{T} \mathbb{1}\left[||f_{i,t} - f_{i,t-1}||_1 < \lambda\right] \sum_{k=1}^{K} |g_\phi(f)_{i,t}^k - g_\phi(f)_{i,t-1}^k|, \quad (7)
$$

The threshold $\lambda$ allows us to discard occlusion areas. The loss function is thus defined by:

$$
\sum_{f\in\mathcal{D}} \mathcal{L}(f,\phi,\theta) = \sum_{f\in\mathcal{D}} \mathcal{L}_r(f,\phi,\theta) + \beta\mathcal{L}_c(f,\phi). \quad (8)
$$

We set $\beta = 1$. The training procedure alternatively updates $\theta$ and $\phi$ with $\alpha$ as learning rate:

$$
\text{for } f \in D: \quad \theta^* = \arg\min_{\theta} L(f,\phi,\theta)(f) \ ; \ \phi_{t+1} = \phi_t - \alpha\nabla_\phi L(f,\phi,\theta^*) \quad (9)
$$

### 3.3 NETWORK ARCHITECTURE

The overall architecture of our unsupervised multiple motion segmentation framework is illustrated in Fig.1. It includes two main modules. The first one, taking the flow volume as input, is a 3D U-net (29). The latent code augmented with embedding positions is directed to the transformer decoder. Then, by cross-attention, the output formed by the volume of segmentation masks is produced. The training of the overall architecture is based on the minimization of the loss function defined below, while the motion model parameters of the segments are given by the B-spline estimation. Temporal consistency is provided by the loss function and the space-time motion models.

## 4 IMPLEMENTATION

### 4.1 IMPLEMENTATION DETAILS

Following (6; 16; 21; 22; 35), we adopt the RAFT method (31) to compute the optical flow fields. More specifically, we use the RAFT version trained on the MPI Sintel dataset (2). We downsample

the computed flow fields to feed the network with $128 \times 224$ vector fields. The output segmentation maps are upsampled to the initial frame size for evaluation. Thus, we can perform efficient training and inference stages. We typically take flow volumes of temporal length $T = 9$ at training time. However, at test time, we can process flow volumes of larger temporal length, and even the flow volume of the whole sequence. To compute of the spatio-temporal parametric motion models, $x$ and $y$ coordinates are normalized within $[-1, 1]$, and a similar normalization is applied to the $t$ coordinate. The full 12-parameter quadratic motion model is chosen in all the experiments, and we take $\nu = 3$ for the frequency factor in the B-spline approximation. We select for each site $i$ the segment $\hat{k}$ with the highest prediction. In eq.(7), we set $\lambda$ as the $99^{th}$ quantile of the flow differences over the flow volume.

Our LT-MS method is very efficient at test time. The computational time amounts on average to 210 *fps* on a GPU P100, that is, twice as fast as the ST-MS method (22). It is certainly due to the long flow sequence given as input to the network, which allows for parallelisation of some heavy computations. In addition, our LT-MS architecture remains lightweight, since it combines only three Unet layers and a transformer decoder on the downsampled feature space. Let us also stress that our LT-MS method does not involve any post-processing at all.

## 4.2 DATA AUGMENTATION AND NETWORK TRAINING

We applied two types of data augmentation dedicated to optical flow. First, we add a global flow to the input flow similarly as in the EM (21) and ST-MS (22) methods. However, in our case, the global flow is given by a full spline-based spatio-temporal motion model whose parameters are chosen at random. The same global flow is added to the flow fields of a given input volume. It allows us to mimic diverse camera motions, enforcing that the motion segments are independent of it. In addition, we corrupt a few flows out of the nine input ones. Thus, we simulate poorly estimated flow fields at some time instants, and the temporal consistency should overcome it.

Our motion segmentation method is fully unsupervised. We never use any manual annotation. We train our model only with the FlyingThings3D (FT3D) dataset (20), whatever the dataset considered at test time. This ensures that our LT-MS network generalizes well to unseen datasets. Regarding hyperparameter setting, we select the stopping epoch from the loss function evaluated on the DAVIS2016 training set. Additional information on the optimization stage is given in the Appendix.

## 5 EXPERIMENTAL RESULTS

We have carried out comparative experiments on four datasets: DAVIS2016[1] (26), SegTrackV2[2] (15), FBMS59 (24), and DAVIS2017-motion (33). More information on the datasets is provided in the Appendix.

### 5.1 ABLATION STUDY

We have conducted an ablation study to assess three main components of our method LT-MS with four masks ($K = 4$), in particular related to the temporal dimension of the problem. We have changed one component at a time as specified hereafter. We use the polynomial space-time quadratic motion model of ST-MS (22) instead of the space-time motion model based on B-splines over the input sequence. We omit the consistency term $\mathcal{L}_c$ in the loss function. We just take the convnet without the transformer decoder. All the ablation experiments were run on the three datasets, DAVIS2016, FBMS59, SegTrackV2. Results are collected in Table 1. In addition, we performed them for two input sequence configurations, respectively input sequences of ten flows, and input sequence of 120 flows (in practice, the whole video for the DAVIS2016 dataset). We can observe that the three ablations have almost the same impact on the performance. The three corresponding model components, i.e., the spline-based motion model, the temporal-consistency loss term, and the transformer decoder are thus all beneficial in similar proportions. They are able to handle the temporal dimension of the problem and the temporal motion evolution along the sequence in a compelling way. Admittedly, the contributions of these three components are more significant for the FBMS59 and the SegTrackV2

---

[1]https://davischallenge.org/index.html

[2]https://paperswithcode.com/dataset/segtrack-v2-1

Table 1: Ablation study for three main components of our method LT-MS ($K = 4$) on DAVIS2016, FBMS59 and SegTrackV2. Only one model component is modified at a time. The performance scores are given by the Jaccard index $\mathcal{J}$. We report ablation results with two input-flow sequence length (or cut size), respectively, by dividing the video into pieces of ten successive frames or by considering 120 successive frames (in practice, the whole video for DAVIS2016).

| Ablation / Dataset | DAVIS2016 | | FBMS59 | | SegTrackv2 | |
|---|---|---|---|---|---|---|
| Cut Size | 10 | 120 | 10 | 120 | 10 | 120 |
| Full Model LT-MS-K4 | 74.8 | 72.4 | 61.0 | 58.2 | 61.3 | 60.4 |
| Unet3D only | 73.0 | 71.3 | 56.6 | 55.5 | 58.2 | 57.3 |
| No consistency term $\mathcal{L}_c$ | 73.5 | 71.0 | 57.5 | 55.5 | 58.0 | 57.5 |
| Polynomial space-time quadratic model | 73.4 | 69.8 | 57.4 | 54.5 | 57.8 | 56.6 |

datasets. However, the dynamic content of the majority of the DAVIS2016 videos, and then, the overall performance score, cannot allow us to fully appreciate the contributions of these three model components. Yet, they can be acknowledged by visualizing results obtained on some videos of DAVIS2016, as shown in the Appendix.

## 5.2 QUANTITATIVE AND COMPARATIVE EVALUATION

We report in Table 2 the results obtained by two versions of our LT-MS method on the three datasets DAVIS2016, SegTrackV2, and FBMS59. LT-MS-K2 performs segmentation with only two masks ($K = 2$) while LT-MS-K4 involves four masks ($K = 4$). Table 2 collects results obtained by our LT-MS method and other existing unsupervised methods, when available. We follow the categorization proposed in (21) regarding input and training. However, we have added a category w.r.t. the network input for four recent methods, (6; 8; 16; 17), that only use RGB images as imput at test time, the optical flow being only involved in the loss function. Evaluation is performed on the binary ground truth (foreground moving object *vs* background) for the three datasets. In the Appendix, we explain how we select the segments for the binary evaluation from the multiple motion segments delivered by our LT-MS method. We still put the OCLR method (33) in the category of unsupervised methods, whereas the authors of the DivA method (14) did not. Indeed, OCLR is not fully unsupervised, since it relies on human-annotated sprites to include realistic shapes in the computer-generated data used at training time. We consider the OCLR version taking only optical flow as input. The post-processing added to the CIS method (36), based on Conditional Random Fields (CRF), is an heavy one, which leads most authors to retain only the version without post-processing for a fair comparison.

As shown in Table 2, our LT-MS method provides very convincing results, both for LT-MS-K2 and LT-MS-K4, in the category of unsupervised methods based on optical flow only. OCLR and DivA demonstrate better performance on the SegTrackV2 dataset. However, as aforementioned, OCLR is not a fully unsupervised method, while DivA leverages RGB images in its conditional decoder. In addition, DivA, along with MoSeg and CIS methods, takes multi-step flows as input between $t$ and in turn $t+1$, $t+2$, $t-1$, $t-2$, and averages the four corresponding predictions to get the final result. SegTrackV2 includes sequences acquired with a poorly controlled handheld camera, which leads to unstable sequences where the contribution of our method is therefore less likely to be emphasized.

Overall, temporal consistency is properly handled over long temporal periods by our LT-MS method. Beyond segmentation performance, we want to stress that our method is the only one providing *by design* a coherent segmentation over the sequence, which is a significant added value. Thus, we can claim that we have not only segmented the moving objects throughout the sequence, but also achieved some kind of tracking.

## 5.3 MULTI-SEGMENT EVALUATION

We have also performed the evaluation of multiple motion segmentation for a multi-segment setting. Since multiple-motion segmentation is harder than the binary motion segmentation (moving foreground *vs* background), accuracy scores are expected to decrease for all methods. In Table 3, we report comparative results on the DAVIS2017-motion dataset and on FBMS59. As for the other methods, we performed segmentation with three masks. To this end, we finetuned the LT-MS-K4

Table 2: Results obtained with two versions of our LT-MS method on DAVIS2016, SegTrackV2, FBMS59. LT-MS-K2 performs segmentation with two masks ($K = 2$), and LT-MS-K4 involves four masks ($K = 4$). We include comparison with unsupervised methods (scores from cited articles). All scores corresponds to evaluation on the binary ground-truth. For LT-MS-K4 and LT-MS-K2, we report results obtained with a cut size of 10. Results with a cut size of 120 are given for LT-MS-K4 in Table 1 with very close performance (additional results on this point in the Appendix). The Jaccard index $\mathcal{J}$ is the intersection over union between the extracted segments and the ground truth, while $\mathcal{F}$ focuses on segment boundary accuracy. Performance is assessed by the average score over all samples, for all datasets but DAVIS2016. For the latter, the overall score is given by the average of sequence scores. *Actually, putting OCLR as an unsupervised method can be questionable (see main text). †DivA somehow uses RGB input since its conditional decoder leverages input images.

| Method | Training | Input | DAVIS2016 $\mathcal{J} \uparrow$ | $\mathcal{F} \uparrow$ | SegTrack V2 $\mathcal{J} \uparrow$ | FBMS59 $\mathcal{J} \uparrow$ |
|---|---|---|---|---|---|---|
| **Ours LT-MS-K4** | Unsupervised | Flow | 74.8 | 72.2 | 61.3 | 61.0 |
| **Ours LT-MS-K2** | | | 70.3 | 68.5 | 58.6 | 55.3 |
| ST-MS (4 masks) (22) | | | 73.2 | 70.3 | 55.0 | 59.4 |
| EM (21) | | | 69.3 | 70.7 | 55.5 | 57.8 |
| MoSeg (35) | | | 68.3 | 66.1 | 58.6 | 53.1 |
| FTS (25) | | | 55.8 | - | 47.8 | 47.7 |
| TIS$_0$ (10) | | | 56.2 | 45.6 | - | - |
| OCLR* (33) (flow only) | | | 72.1 | - | 67.6 | 65.4 |
| GWM (6) | | RGB (Flow in loss) | 79.5 | - | 78.3 | 77.4 |
| RCF (16) | | | 80.9 | - | 76.7 | 69.9 |
| AMD (17) | | | 57.8 | - | 57.0 | 47.5 |
| MOD (8) | | | 73.9 | - | 62.2 | 61.3 |
| DivA(4)† (14) | | RGB & Flow | 72.4 | - | 64.6 | - |
| TIS$_s$ (10) | | | 62.6 | 59.6 | - | - |
| CIS - No Post (36) | | | 59.2 | - | 45.6 | 36.8 |
| CIS - With Post (36) | | | 71.5 | - | 62.0 | 63.6 |

Table 3: Multi-segment evaluation. Regarding DAVIS2017-motion, $\mathcal{J}\&\mathcal{F}$ is the mean of the two. Evaluation is performed on the video as a whole according to the official DAVIS2017 scheme. Reported scores are the average of the individual video scores. *See caption of Table 2. For FBMS59, the two evaluation metrics are bootstrap IoU (bIoU) (14) where each ground-truth mask is mapped to the most likely predicted segment (performed at the frame level), and linear assignment that is a bilinear mapping between the ground-truth and the predicted segments at the sequence level (similar to the official DAVIS2017 evaluation).

| Dataset | DAVIS2017-motion | | | FBMS59 | |
|---|---|---|---|---|---|
| Method / Scores | $\mathcal{J}\&\mathcal{F} \uparrow$ | $\mathcal{J} \uparrow$ | $\mathcal{F} \uparrow$ | bIoU | Linear Assignment |
| **Ours LT-MS-K3** | 42.2 | 39.3 | 45.0 | 58.4 | 47.2 |
| ST-MS (22) | 42.0 | 38.8 | 45.2 | - | - |
| MoSeg (35) | 35.8 | 38.4 | 33.2 | - | - |
| OCLR* (33) | 55.1 | 54.5 | 55.7 | - | - |
| DivA (14) | - | - | - | 42.0 | - |

network on the DAVIS2016 training set with now three masks ($K = 3$). The resulting performance on DAVIS2017-motion is slightly better than ST-MS (22), and far better than MoSeg. There is still the same remark about OCLR status as an unsupervised method. Regarding FBMS59, we report multimask segmentation results with two different metrics. Our LT-MS method outperforms by a large margin DivA (14) that attempted this multi-segment evaluation on FBMS59.

## 5.4 QUALITATIVE VISUAL EVALUATION

Fig.3 contains several visual results to demonstrate how our LT-MS method behaves on different situations. We display three result samples obtained on different videos of the benchmarks. Additional results are provided in the Appendix. We can observe that the motion segments are globally accurate. Since our method involves several masks, we can properly handle articulated motions (*people2, bmx*), deal with the presence of several moving objects in the scene (*people2*), separate

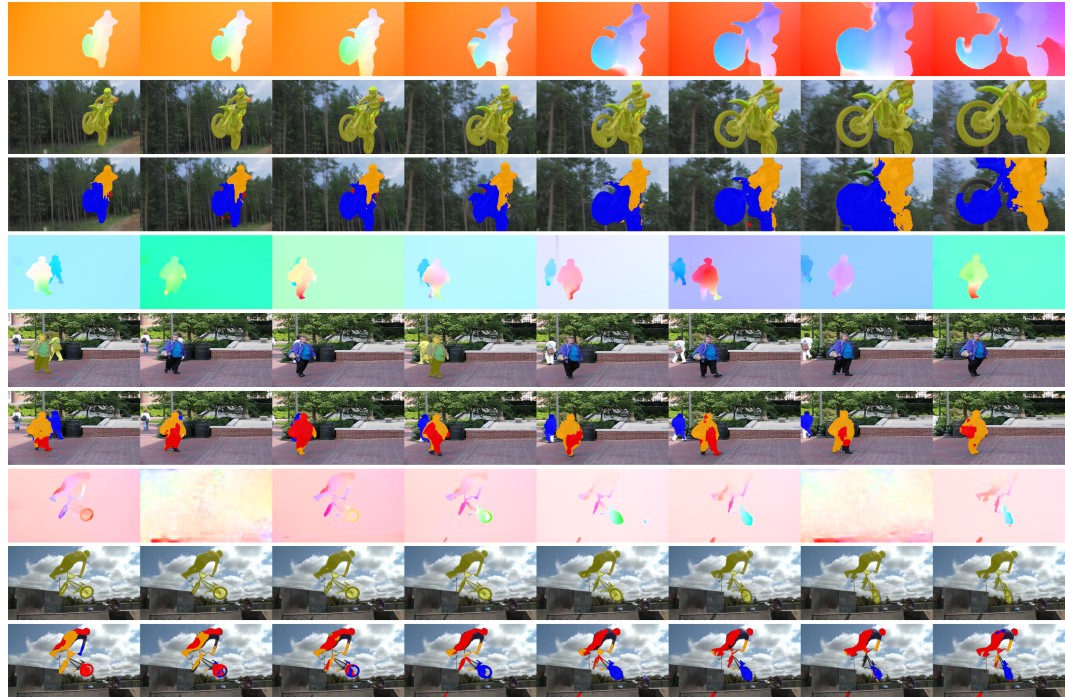

Figure 3: Results obtained with our LT-MS-K4 method ($K = 4$). Three groups of results are displayed, *motocross-jump* from DAVIS2016, *people02* from FBMS59 and *bmx* from SegTrackV2. For each group, the first row samples successive flow fields (HSV color code) corresponding to the processed video. The second row contains the corresponding images of the video, where the ground-truth of the moving object is overlaid in yellow (when available at that frame). The third row shows the motion segments provided by our LT-MS-K4 method with one colour per segment. For all the results, we adopt the same color set for the three masks corresponding to the moving objects (blue, red and orange), and we let the background image for the background mask.

the rider from the bike or motorbike (*bmx, motocross-jump*), or accomodate motion parallax. Since our method enforces temporal consistency, it can also deal with errors in optical flow estimation or with objects that momentarily stop moving (*bmx*). We must keep in mind that our actual target is the optical-flow segmentation (OFS) task, even if we evaluate our method on VOS benchmarks. When VOS benchmarks deal with the segmentation of one primary object moving in the foreground, it may occur discrepancies with OFS, which negatively impacts the evaluation scores. The segmentation of additional parts, which appears wrong w.r.t. VOS ground truth, on the contrary makes sense from the OFS standpoint.

## 6 Conclusion

We have designed an original transformer-based unsupervised method for segmenting multiple motions over a video in one go. Our LT-MS method leverages the ELBO framework for the loss function, and fully acknowledges the temporal dimension of the motion segmentation problem. Indeed, to the best of our knowledge, our method is the first unsupervised network-based OFS method explicitly leading to a stable and consistent OF segmentation throughout long video sequences. It introduces at training time, on one hand, B-splines spatio-temporal parametric motion models over space-time volumes, and on the other hand, a loss term expressing temporal consistency over successive masks while taking care of occlusions. Our transformer-based network can be applied at test time to input volumes of any time length. It can accomodate different choices on the number $K$ of masks with a simple finetuning step. Experimental results on several datasets demonstrate the efficiency and the accuracy of our LT-MS method by providing competitive results on several datasets. In addition, it is very fast at test time. In future work, we will investigate the slot attention mechanism to modify the number of masks at test time without retraining the model.

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
