# Unsupervised motion segmentation in one go: Smooth long-term model over a video

## A    Appendix

### A.1    Loss function

Here, we motivate the expression of the temporal consistency term $\mathcal{L}_c$ of the loss function, given by Eq.7 in the main text.

First, let us recall the main definitions and notations. We consider a volume of optical flow fields $f \in \mathbb{R}^{2 \times T \times W \times H}$, the spatial grid $\Omega \in \mathbb{R}^{W \times H}$ and $T$ temporal steps. We assume that we can decompose the flow as a set of $K$ segments, each one exhibiting a coherent motion. Flow vectors within a given segment $k$ can be represented by a smooth parametric motion model parametrized by $\theta_k$. $\theta = \{\theta_k, k = 1 \cdots K\}$. Variable $z_{i,t}$ conveys the motion segmentation, $z_{i,t}^k = 1$ if site $(i,t)$ belongs to segment $k$ ($z_{i,t}^k = 0$ otherwise). $z$ and $\theta$ are latent variables, and $f$ is the observed data.

#### A.1.1    Prior

We introduce a prior on $z$:

$$\forall i \in I, \forall t \in \{1 \cdots T\}, z_{i,t} \in \{1 \cdots K\}$$

$$p(z) = \prod_i p(z_{i,1}) \prod_{t=2}^{T} p(z_{i,t}|z_{i,t-1})$$

$$p(z_{i,t} = k|z_{i,t-1} = l) = \frac{1}{M} \exp(\alpha \mathbb{1}[k = l] + \beta \mathbb{1}[k \neq l])$$

$$p(z_{i,1} = k) = \frac{1}{K}.$$

We also use the notation $g_{i,t}^k = g_\phi(f)_{i,t}^k = q(z_{i,t} = k|f)$.

Using an uninformative prior on $\theta$, we can compute the KL term as:

$$KL[q(z,\theta|f)||p(z,\theta)] = KL[q(z|f)||p(z)] \tag{1}$$

$$= \mathbb{E}_q[\log q(z|f)] - \mathbb{E}_q[\log p(z)] \tag{2}$$

$$\mathbb{E}_q[\log q(z|f)] = \sum_{i,t,k} g_{i,t}^k \log g_{i,t}^k \tag{3}$$

$$\mathbb{E}_q[\log p(z)] = \sum_i [\mathbb{E}_q[\log p(z_{i,1})] + \sum_{t=2}^{T} \mathbb{E}_q[\log p(z_{i,t}|z_{i,t-1})]] \tag{4}$$

$$= \sum_i \sum_{t=2}^{T} \mathbb{E}_q[\log p(z_{i,t}|z_{i,t-1})] - I \log(K) \tag{5}$$

$$= \sum_i \sum_{t=2}^{T} \sum_k^K \sum_l^L g_{i,t}^k g_{i,t-1}^l (\alpha \mathbb{1}[k = l] + \beta \mathbb{1}[k \neq l]) + c, \tag{6}$$

where $c = -I(\log(K) + T \log(M))$ is a constant.

We set $\beta = 0$, knowing that $\alpha > 0$ :

$$\mathbb{E}_q[\log p(z)] = \alpha \sum_i \sum_{t=2}^{T} \sum_k^{K} g_{i,t}^k g_{i,t-1}^k + c. \tag{7}$$

### A.1.2  APPROXIMATION

$$2 * g_{i,t}^k g_{i,t-1}^k = -(g_{i,t}^k - g_{i,t-1}^k)^2 + (g_{i,t}^k)^2 + (g_{i,t-1}^k)^2 \tag{8}$$

Thus, we can write:

$$\mathbb{E}_q[\log p(z)] = \frac{\alpha}{2} \sum_i \sum_{t=2}^{T} \sum_k^{K} -(g_{i,t}^k - g_{i,t-1}^k)^2 + (g_{i,t}^k)^2 + (g_{i,t-1}^k)^2 + c. \tag{9}$$

$$\mathbb{E}_q[\log p(z)] = \frac{\alpha}{2} (\sum_i \sum_{t=2}^{T} \sum_k^{K} -(g_{i,t}^k - g_{i,t-1}^k)^2$$
$$+ \sum_i \sum_{t=1}^{T} \sum_k^{K} (g_{i,t}^k)^2 + \sum_i \sum_{t=2}^{T-1} \sum_k^{K} (g_{i,t}^k)^2) \tag{10}$$

Thus, we get:

$$KL[q(z|f)||p(z)] = \frac{\alpha}{2} \sum_i \sum_{t=2}^{T} \sum_k (g_{i,t-1}^k - g_{i,t-1}^k)^2 + \sum_i \sum_{t=1}^{T} \sum_k g_{i,t}^k * (\log(g_{i,t}^k)$$
$$- \frac{\alpha}{2} * g_{i,t}^k) - \frac{\alpha}{2} \sum_i \sum_{t=2}^{T-1} \sum_k^{K} (g_{i,t}^k)^2) + c. \tag{11}$$

$\forall x \in [0,1]; \alpha > 0 : -\frac{\alpha}{2} \leq x(\log x - \frac{\alpha}{2}x) \leq 0$, therefore:

$$KL[q(z|f)||p(z)] \leq \frac{\alpha}{2} \sum_{i,t,k} (g_{i,t}^k - g_{i,t-1}^k)^2 + c$$

### A.1.3  COMBINING WITH LIKELIHOOD

$$Elbo(f) = \mathbb{E}_{q(z,\theta|f)}[\log p(f|z,\theta)] - KL(q(z,\theta|f)||p(z,\theta)) - c$$

$$Elbo(f) \geq \mathbb{E}_{q(z,\theta|f)}[\log p(f|z,\theta)] - \frac{\alpha}{2} \sum_i \sum_{t=2}^{T} \sum_k (g_{i,t}^k - g_{i,t-1}^k)^2 - c$$

As $\forall x, y \in [0,1]^2 : |x - y| \geq (x - y)^2$:

$$Elbo(f) \geq \mathbb{E}_{q(z,\theta|f)}[\log p(f|z,\theta)] - \frac{\alpha}{2} \sum_i \sum_{t=2}^{T} \sum_k |g_{i,t}^k - g_{i,t-1}^k| - c \tag{12}$$

$$Elbo(f) \geq - \sum_{t=1}^{T} \sum_{i \in \Omega} \sum_k g_\phi(f)_{i,t}^k ||f(i,t) - \tilde{f}_{S_n(\theta_k)}(i,t)||_1 / \xi_{f,t} \tag{13}$$

$$- \frac{\alpha}{2} \sum_{i,t=2,k} |g_{i,t}^k - g_{i,t-1}^k| - c \tag{14}$$

This is the lower bound we are maximising, although, in practice we are not accounting a fraction of the sites to deal with occlusion areas.

### A.1.4  EXPERIMENTAL VALIDATION

In order to evaluate how similar the objective we trained in the paper is to the expression defined in Eq.7 above, we train a model minimizing the following loss function:

Table 1: Comparison of the two losses

| Prior / Dataset | DAVIS2016 | | FBMS59 | | SegTrackv2 | |
|---|---|---|---|---|---|---|
| **Cut Size** | 10 | 120 | 10 | 120 | 10 | 120 |
| Full Model LT-MS-K4 | 74.8 | 72.4 | 61.0 | 58.2 | 61.3 | 60.4 |
| Prior Eq.7 | 73.5 | 71.6 | 57.6 | 55.6 | 59.0 | 57.4 |

$$\mathcal{L} = \sum_{t=1}^{T} \sum_{i \in \Omega} \sum_{k} g_\phi(f)_{i,t}^k ||f(i,t) - \tilde{f}_{S_n(\theta_k)}(i,t)||_1 / \xi_{f,t}$$

$$- \alpha \sum_{i} \sum_{t=2}^{T} \sum_{k}^{K} g_{i,t}^k g_{i,t-1}^k + \beta \sum_{t=1}^{T} \sum_{i \in \Omega} \sum_{k} g_\phi(f)_{i,t}^k \log g_\phi(f)_{i,t}^k \qquad (15)$$

In practice, we use $\alpha = 0.5$ and $\beta = 0.01$, and we obtain the results reported in Table 1.

## A.2 DATASETS

DAVIS2016 comprises 50 videos (for a total of 3455 frames), depicting diverse moving objects. It is split in a training set of 30 videos and a validation set of 20 videos. Only the primary moving object is annotated in the ground truth. The criteria for evaluation on this dataset are the Jaccard score $\mathcal{J}$ (intersection-over-union), and the contour accuracy score $\mathcal{F}$, the higher, the better for both. FBMS59 includes 59 videos (720 annotated frames), and SegTrackV2 14 videos (1066 annotated frames). Both mostly involve one foreground moving object, but sometimes a couple of moving objects. For FBMS59, we use the 30 sequences of the validation set. Although annotations may comprise multiple objects, the VOS community exploits FBMS59 and SegTrackV2 similarily as DAVIS2016, considering a binary ground-truth, by grouping moving objects in the foreground. We follow this practice. However, we also provide a multi-mask evaluation for FBMS59.

DAVIS2017 is an extension of DAVIS2016, containing 90 videos. It includes additional videos with multiple moving objects, and it provides multiple-segment annotations for the ground truth. It is split into 60 videos for training and 30 for evaluation. DAVIS2017-motion is a curated version of the DAVIS2017 dataset proposed in (5) for a fair evaluation of methods based on optical flow only. Connected objects exhibiting the same motion are merged in the ground truth for evaluation. We proceed to the evaluation with the official algorithm that involves a Hungarian matching process.

## A.3 ADDITION TO THE ABLATION STUDY

Fig.1 plots the performance scores for the three ablations when the length of the input flow sequence varies from 10 to 120. Overall, they exhibit comparable behaviour at a certain distance from the full model.

Visual results reported in Fig.2 clearly demonstrate that the addition of the temporal consistency loss term $\mathcal{L}_c$ allows us to get far more consistent segments over time, whether for the background, or the moving objects. For instance, in the fourth example, the foreground moving car is perfectly segmented throughout the sequence along with its wheels, and (small) moving cars in the background. Fig.5 highlights the contribution of the spline-based motion model on the *dog* video, and its obvious ability to handle motions that do not vary uniformly, as the erratic movement of the dog.

## A.4 OPTIMIZATION IMPLEMENTATION

We use Adam optimizer with the following strategy on the learning rate $\alpha$ to train the network (3D Unet and transformer), inspired from the warmup-decay strategy in (5). We linearly increase it from 0 to $1e - 4$ for 20 epochs, then, we divide it by two every 40 epochs. The estimation of the motion model parameters through the B-spline approximation at training time is achieved with the Pytorch implementation of L-BGFS (3).

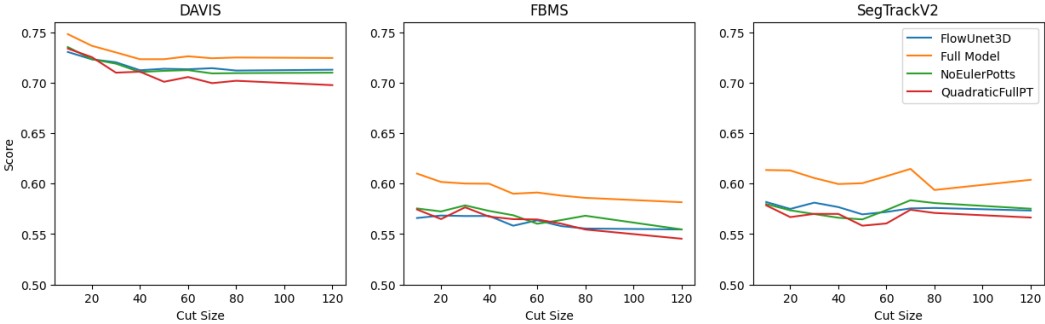

Figure 1: Influence of the length of the input flow sequence (also referred to as cut size on the horizontal axis of the plot) on the three modified versions of our LT-MS-K4 method with comparison with the full model. Left plot: for the DAVIS2016 dataset. Middle plot: for the FBMS59 dataset. Right plot: for the SegTrackV2 dataset. Overall, they exhibit comparable behaviour at a certain distance from the full model.

## A.5   SEGMENT SELECTION FOR EVALUATION

VOS and OFS are close but not identical tasks. However, we use VOS benchmarks, since no OFS benchmarks are available. Let us recall that the VOS one is attached to the notion of a primary object of interest moving in the foreground (sometimes, a couple of objects). As a consequence, we have to select the right segments to cope with the binary ground truth of the VOS benchmarks, as usually done for the DAVIS2016, SegTrackV2 and FBMS59 datasets.

Since we deal with multiple-motion segmentation, i.e., $K$ segments, we have to group them into two clusters corresponding to the foreground moving object on one side and the background on the other side. We proceed as in the ST-MS method (4). However, in our case, we can directly evaluate our LT-MS method in one shot over the predicted segmentation sequence, since there is no temporal linking postprocessing for LT-MS. For evaluations with $K = 2$, we do not need to group the masks, so we just set one of the masks as foreground using Hungarian matching over the sequence (alternatively, we could just pick the smallest of the two masks). Regarding the evaluation on the DAVIS2017-motion dataset whose ground truth is not binary, we use the official evaluation script (which perform hungarian matching over the sequence) to associate each predicted segment $k$ with the ground-truth annotations, knowing that we have to consider $K = 3$ masks for this experimentation.

## A.6   MULTI-MASK EVALUATION IN FBMS59

We implement the "Boostrap IoU" score to evaluate multi-mask segmentation in FBMS59, as described in (2)[1] The core idea is to match each ground-truth segment to the most likely segment (i.e., with the one of highest IoU). Let us note that this evaluation includes the background segment, since background is not identified in the FBMS59 multi-mask annotation.

The algorithm for evaluation we used is:

```
sequences_iou = []
for sequence in dataset :
    objects_iou = []
    for frame in sequence :
        for x in gt_annotations_including_background :
            maxiou = 0
            for y in predicted_segments :
                maxiou = max(maxiou, iou(x, y))
            objects_iou.append(maxiou)
    sequence_iou.append(mean(objects_iou))
dataset_iou = mean(sequences_iou)
```

---
[1]We use the multi-label ground truth provided by Dong Lao, first author of (2).

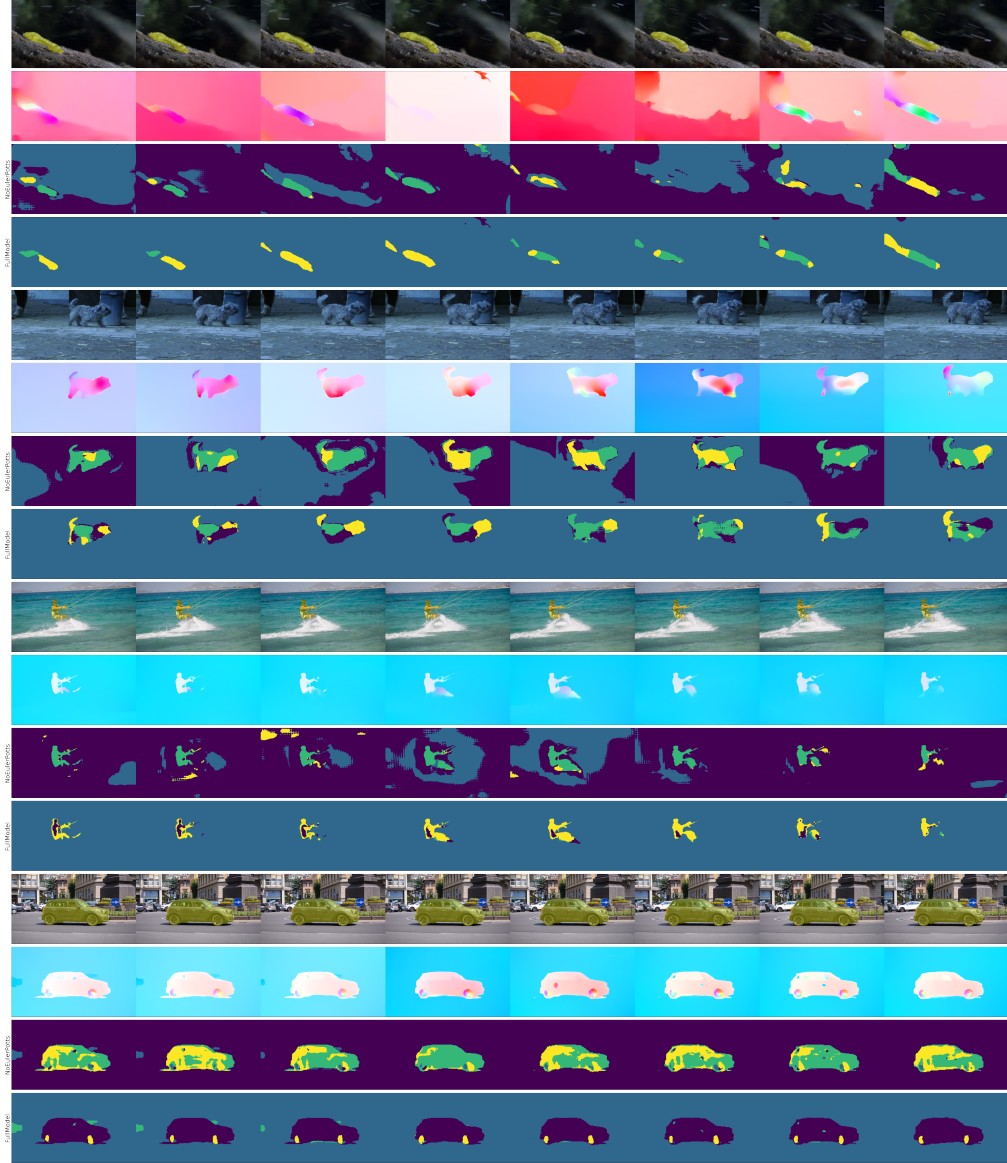

Figure 2: Four groups of qualitative results regarding the ablation of the temporal-consistency loss term. They respectively correspond to the *worm* video of SegTrackV2, the *dogs01* video of FBMS59, and the *kite-surf* and *car-roundabout* videos of DAVIS2016. For each group, the first row contains sample images with the segmentation ground-truth, when available at that frame, overlaid in yellow, the second row displays the input flows, the third and fourth rows show the predicted motion segmentations, respectively without and with the temporal consistency loss term. Clearly, this model component allows us to get far more consistent segments over time.

For the linear assignment score, we find the best bipartite sequence-level match between the ground truth and the predictions. Similar to the bIoU evaluation, we compute the score with all the labels of the ground truth. This evaluation is more demanding, because it forces a one-to-one match between the prediction and the masks at the sequence level.

## A.7 REPEATABILITY

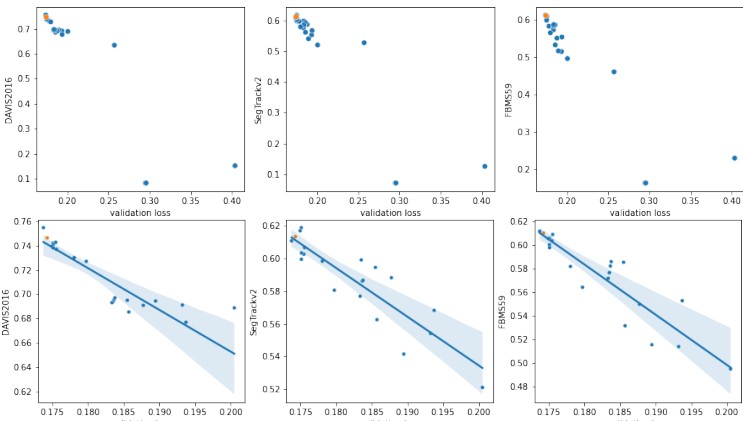

Figure 3: Model scores depending on initialization. Each dot represents a trained model. The $x$-axis represents the validation loss on the DAVIS 2016 train dataset, and the $y$-axis the performance on the evaluation dataset. The top row includes all models and the bottom row excludes models that diverged during training (validation loss $> 0.25$). The bottom row displays the linear relationship between validation loss and evaluation score. The model whose results are reported in the main text is represented as an orange dot.

With the introduction of the transformer decoder, we experimentally found that the convergence of the network depends on the weights initialization, and that the same network and loss configuration can yield different results at test time. In Fig.3, we show that our unsupervised loss on a held-out validation set is a good indicator of the network performance at test time. This is a critical point for model and hyperparameter selection, since we do not have access to the ground truth at training time (with our fully unsupervised scenario), and thus we cannot evaluate model performance. Fig.4 plots the model performance after training as a function of the initialization budget.

In our evaluation experiments, we account for this randomness by training five models with the same set of seeds for all ablations and by reporting the score of the model with the lowest validation loss for each model.

## A.8 TRAINING ON REAL-WORLD DATA

Table 2: Training on synthetic (FT3D) and on real (DAVIS2016) data

| Training mode / Dataset | DAVIS2016 | | FBMS59 | | SegTrackv2 | |
|---|---|---|---|---|---|---|
| **Cut Size** | 10 | 120 | 10 | 120 | 10 | 120 |
| Full Model LT-MS-K4 (FT3D) | 74.8 | 72.4 | 61.0 | 58.2 | 61.3 | 60.4 |
| Full Model LT-MS-K4 (DAVIS2016) | 73.6 | 72.8 | 60.0 | 57.6 | 62.8 | 61.3 |

We have trained our model LT-MS-K4 on the DAVIS2016 training set and evaluated it on DAVIS2016 validation set, FBMS59 and SegTrackV2. Results are collected in Table 2 and compared with those obtained when training our model on the synthetic FT3D dataset. We can observe that performance when training on DAVIS2016 remains globally on par with performance when training on FT3D. Let us note that the volume of data is much smaller when training on DAVIS 2016.

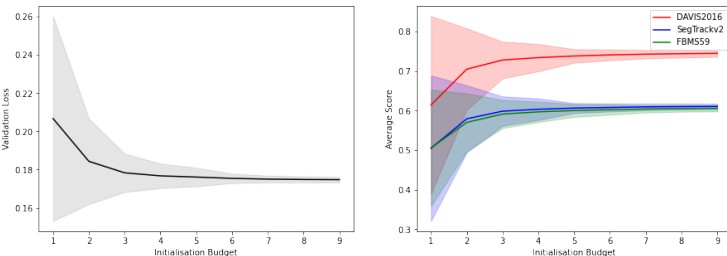

Figure 4: Evolution of the model performance as a function of the initialization budget. Left : the validation loss associated to the best network of each subset. Right : average performance on the test data set associated to this network for each different budget. The filled area correspond to +/- the standard deviation for each curve.

## A.9    INFLUENCE OF THE CUT SIZE AT INFERENCE

We have also tested how our LT-MS-K4 method behaves when varying the length of the input flow sequence. Results are plotted in Fig.6 for three datasets, DAVIS2016, FBMS59 and SegTrackV2. As expected, the best performance is obtained for the smallest length (equal to 10) of the input flow sequence. However, performance decreases slowly when the length increases, and remains stable for larger ones. It demonstrates the intrinsic ability of the LT-MS method to achieve accurate and consistent motion segmentation over long periods of the video, which is a unique property. We also did it for LT-MS-K2 as reported in Table 3. We even get slightly better results for DAVIS2016 and SegTrackV2 when the video sequence is processed in one go, i.e., with an infinite cutsize.

Table 3: Results obtained on the three datasets DAVIS2016, FBMS59 and SegTrackV2 with LT-MS-K2 for respectively a cut size of 10 and no cut (the video sequence is processed in one go).

| Dataset  | DAVIS2016 | | FBMS59 | | SegTrackV2 | |
|----------|-----------|----------|--------|----------|------------|----------|
| Cut Size | 10 | $\infty$ | 10 | $\infty$ | 10 | $\infty$ |
| LT-MS-K2 | 70.3 | 70.7 | 55.3 | 48.7 | 58.6 | 59.3 |

## A.10    ADDITIONAL VISUAL EVALUATION

Additional visual results of our LT-MS method are provided in Fig.7. We can observe that the motion segments are globally accurate. Let us recall that the ground-truth is not necessarily available for all the video frames depending on the datasets. Since our method involves several masks, we can properly handle articulated motions (*monkey*), deal with the presence of several moving objects in the scene (*hummingbird*), or accomodate motion parallax (*libby*).

In Fig.8, we collect additional visual results on three videos of the SegTrackV2 dataset, from top to bottom, the *birdfall*, *bird-of-paradise*, and *bmx* videos. Results demontrate the ability of our long-term segmentation method to recover, to some extent, correct segmentation even from flow fields wrongly computed at some time instants.

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

## A.11 DETAILED RESULTS PER VIDEOS OF THE DATASETS

Hereafter, we report detailed results through tables collecting the evaluation scores obtained by our LT-MS-K4 method for every video of the two datasets DAVIS2016 and Seg-TrackV2 (binary evaluation), and by our LT-MS-K3 model for every video of the DAVIS2017-motion dataset and FBMS59 (multi-mask evaluation). Let us recall that the official evaluation algorithm is not the same for DAVIS2016 and DAVIS2017-motion. The evaluation is done on the whole video for DAVIS2017-motion and is multi-segment, while it is binary and performed frame by frame of the video for DAVIS2016.

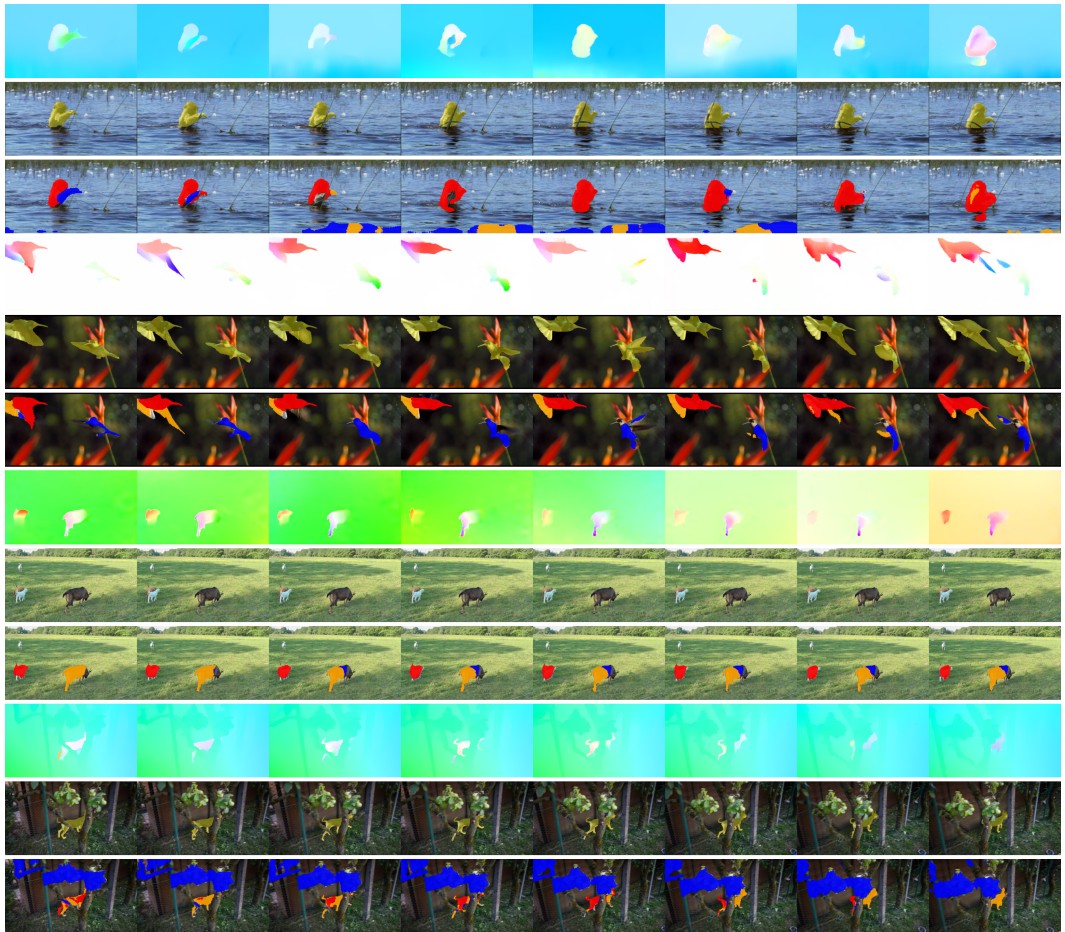

Figure 7: Results obtained with our LT-MS-K4 method ($K = 4$). Four groups of results are displayed: *monkey*, *hummingbird* from SegTrackV2, *goats01* from FBMS59 and *libby* from DAVIS2016. For each group, the first row samples successive flow fields (HSV color code) corresponding to the processed video. The second row contains the corresponding images of the video, where the ground-truth of the moving object is overlaid in yellow (when available at that frame). The third row shows the motion segments provided by our LT-MS-K4 method with one colour per segment. For all the results, we adopt the same color set for the three masks corresponding to the moving objects (blue, red and orange), and we let the background image for the background mask.

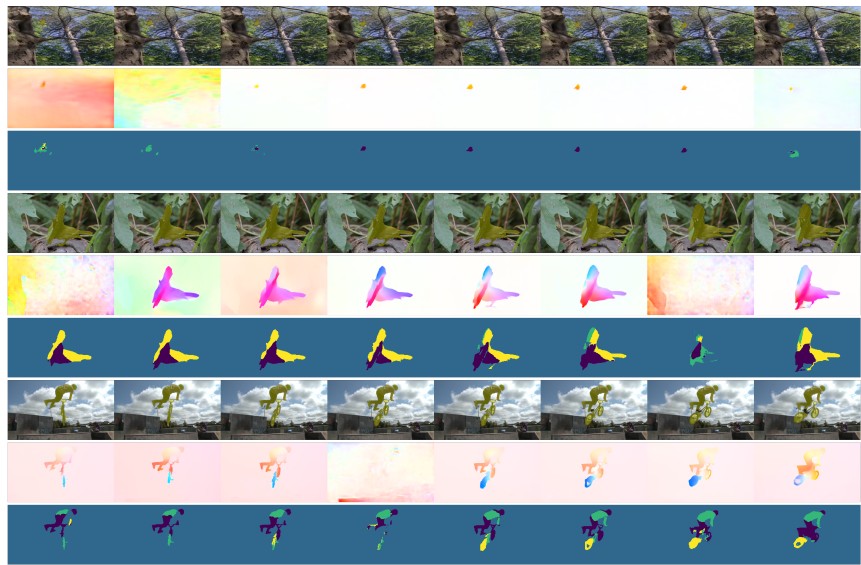

Figure 8: Illustration of the temporal consistency provided by our LT-MS-K4 method for three examples, the *birdfall*, *bird-of-paradise*, and *bmx* videos of SegTrackV2. For each group, the first row contains the video images with the ground truth overlaid in yellow when available; the second row depicts the corresponding flow fields represented with the HSV code while normalized independently from each other; the third row provides the predicted segmentation.

### A.11.1 DAVIS2016

| Video | $\mathcal{J}$ (M) | $\mathcal{J}$ (O) | $\mathcal{J}$ (D) | $\mathcal{F}$ (M) | $\mathcal{F}$ (O) | $\mathcal{F}$ (D) |
|---|---|---|---|---|---|---|
| blackswan | 0.515 | 0.521 | -0.091 | 0.561 | 0.833 | -0.036 |
| bmx-trees | 0.596 | 0.744 | 0.234 | 0.795 | 0.923 | 0.125 |
| breakdance | 0.740 | 1.000 | 0.038 | 0.722 | 1.000 | 0.000 |
| camel | 0.867 | 1.000 | 0.104 | 0.858 | 1.000 | 0.101 |
| car-roundabout | 0.920 | 1.000 | -0.004 | 0.806 | 1.000 | -0.060 |
| car-shadow | 0.901 | 1.000 | 0.011 | 0.848 | 1.000 | -0.017 |
| cows | 0.872 | 1.000 | 0.026 | 0.798 | 1.000 | 0.009 |
| dance-twirl | 0.821 | 1.000 | -0.113 | 0.842 | 1.000 | -0.045 |
| dog | 0.769 | 1.000 | -0.085 | 0.671 | 1.000 | -0.076 |
| drift-chicane | 0.718 | 0.860 | 0.063 | 0.793 | 0.860 | 0.161 |
| drift-straight | 0.892 | 1.000 | 0.047 | 0.838 | 1.000 | 0.229 |
| goat | 0.380 | 0.330 | -0.087 | 0.367 | 0.114 | -0.069 |
| horsejump-high | 0.843 | 1.000 | 0.064 | 0.896 | 1.000 | -0.002 |
| kite-surf | 0.518 | 0.500 | 0.114 | 0.494 | 0.375 | -0.035 |
| libby | 0.781 | 1.000 | 0.108 | 0.892 | 1.000 | 0.035 |
| motocross-jump | 0.750 | 0.842 | 0.042 | 0.590 | 0.658 | 0.092 |
| paragliding-launch | 0.621 | 0.667 | 0.299 | 0.308 | 0.179 | 0.357 |
| parkour | 0.711 | 0.878 | -0.315 | 0.765 | 0.949 | -0.157 |
| scooter-black | 0.854 | 1.000 | -0.036 | 0.729 | 1.000 | 0.072 |
| soapbox | 0.891 | 1.000 | 0.020 | 0.863 | 1.000 | 0.020 |
| Average | 0.748 | 0.867 | 0.022 | 0.722 | 0.845 | 0.035 |

Table 4: Results given for every video of DAVIS2016 dataset. Reported scores per video are the average Jaccard score over frames in the video. The very last row is the average over videos scores. $\mathcal{J}$ is the Jaccard index and $\mathcal{F}$ is the Countour Accuracy. The Mean ($M$) is the average of the scores, the Recall ($O$) is the fraction of frames per video with a score higher than $0.5$, and the Decay ($D$) is the degradation of the score over time in the video.

A.11.2   SEGTRACKV2

| Video | Jacc ($\mathcal{J}$) |
|---|---|
| bird of paradise | 0.596 |
| birdfall | 0.468 |
| bmx | 0.776 |
| cheetah | 0.438 |
| drift | 0.431 |
| frog | 0.792 |
| girl | 0.651 |
| hummingbird | 0.697 |
| monkey | 0.607 |
| monkeydog | 0.241 |
| parachute | 0.928 |
| penguin | 0.517 |
| soldier | 0.752 |
| worm | 0.514 |

Table 5: Results given for every video of SegTrackV2 dataset. Each reported score is the average Jaccard score over annotated frames in the video. The very last row is the average over all the frames and over all the videos.

### A.11.3 DAVIS2017-MOTION

| Sequence | J-Mean | F-Mean |
|---|---|---|
| bike-packing˙1 | 0.078 | 0.327 |
| bike-packing˙2 | 0.254 | 0.267 |
| blackswan˙1 | 0.477 | 0.575 |
| bmx-trees˙1 | 0.608 | 0.804 |
| breakdance˙1 | 0.450 | 0.526 |
| camel˙1 | 0.772 | 0.726 |
| car-roundabout˙1 | 0.913 | 0.810 |
| car-shadow˙1 | 0.896 | 0.845 |
| cows˙1 | 0.821 | 0.728 |
| dance-twirl˙1 | 0.444 | 0.576 |
| dog˙1 | 0.654 | 0.571 |
| dogs-jump˙1 | 0.019 | 0.147 |
| dogs-jump˙2 | 0.254 | 0.323 |
| dogs-jump˙3 | 0.303 | 0.351 |
| drift-chicane˙1 | 0.557 | 0.627 |
| drift-straight˙1 | 0.886 | 0.829 |
| goat˙1 | 0.220 | 0.300 |
| gold-fish˙1 | 0.018 | 0.240 |
| gold-fish˙2 | 0.282 | 0.336 |
| gold-fish˙3 | 0.357 | 0.356 |
| gold-fish˙4 | 0.000 | 0.000 |
| gold-fish˙5 | 0.000 | 0.000 |
| horsejump-high˙1 | 0.721 | 0.794 |
| india˙1 | 0.066 | 0.085 |
| india˙2 | 0.163 | 0.172 |
| india˙3 | 0.072 | 0.108 |
| judo˙1 | 0.225 | 0.397 |
| judo˙2 | 0.286 | 0.414 |
| kite-surf˙1 | 0.449 | 0.464 |
| lab-coat˙1 | 0.332 | 0.350 |
| libby˙1 | 0.746 | 0.853 |
| loading˙1 | 0.057 | 0.246 |
| loading˙2 | 0.128 | 0.259 |
| loading˙3 | 0.427 | 0.514 |
| mbike-trick˙1 | 0.444 | 0.436 |
| motocross-jump˙1 | 0.446 | 0.435 |
| paragliding-launch˙1 | 0.577 | 0.286 |
| parkour˙1 | 0.531 | 0.646 |
| pigs˙1 | 0.105 | 0.433 |
| pigs˙2 | 0.437 | 0.431 |
| pigs˙3 | 0.059 | 0.217 |
| scooter-black˙1 | 0.843 | 0.724 |
| shooting˙1 | 0.434 | 0.568 |
| soapbox˙1 | 0.487 | 0.696 |

Table 6: Results given for every video of D17 dataset.

| J&FMean | JMean | JRecall | JDecay | FMean | FRecall | FDecay |
|---|---|---|---|---|---|---|
| 0.422 | 0.393 | 0.387 | 0.004 | 0.450 | 0.437 | 0.024 |

Table 7: Results given for every video of DAVIS2017-motion dataset. The very last row is the average score over all the videos for the different criteria.

### A.11.4    FBMS59 - MULTIPLE MASKS

| Video | bIoU ($\mathcal{J}$) | Linear Assignment Sequence ($\mathcal{J}$) |
|---|---|---|
| camel01 | 0.599 | 0.591 |
| cars1 | 0.698 | 0.527 |
| cars10 | 0.484 | 0.324 |
| cars4 | 0.78 | 0.561 |
| cars5 | 0.578 | 0.515 |
| cats01 | 0.76 | 0.76 |
| cats03 | 0.698 | 0.492 |
| cats06 | 0.469 | 0.393 |
| dogs01 | 0.814 | 0.814 |
| dogs02 | 0.766 | 0.434 |
| farm01 | 0.613 | 0.519 |
| giraffes01 | 0.502 | 0.446 |
| goats01 | 0.424 | 0.336 |
| horses02 | 0.549 | 0.436 |
| horses04 | 0.458 | 0.366 |
| horses05 | 0.327 | 0.145 |
| lion01 | 0.426 | 0.376 |
| marple12 | 0.386 | 0.324 |
| marple2 | 0.501 | 0.291 |
| marple4 | 0.945 | 0.945 |
| marple6 | 0.446 | 0.311 |
| marple7 | 0.643 | 0.44 |
| marple9 | 0.384 | 0.231 |
| people03 | 0.406 | 0.298 |
| people1 | 0.892 | 0.892 |
| people2 | 0.741 | 0.616 |
| rabbits02 | 0.507 | 0.445 |
| rabbits03 | 0.578 | 0.514 |
| rabbits04 | 0.615 | 0.444 |
| tennis | 0.519 | 0.382 |
| Seq Average | 0.584 | 0.472 |

Table 8: Results given for every video of FBMS59 dataset. Each reported score is the average Jaccard score over annotated frames in the video. The very last row is the average of the score over videos.