# OpenReview forum: "Unsupervised motion segmentation in one go: Smooth long-term model over a video"
_ICLR.cc/2024/Conference — Submitted to ICLR 2024_

### Official Review · Reviewer_po8i · 2023-10-30

**Soundness:** 3 good
**Presentation:** 2 fair
**Contribution:** 3 good
**Rating:** 5
**Confidence:** 4

**Summary:**

The paper proposes a novel method for motion segmentation that, for the loss, combines a 12-parameter quadratic motion model with B-Splines to model long-term temporal consistency and object motion evolution.
The model trains an optical flow segmentation network to receive a stack of flow frames up to potentially the whole video. The method is unsupervised and trained on synthetic FlyingThings3D data. The network is then applied to 4 benchmark datasets (DAVIS 2016, FBMS, SegTrackv2 and DAVIS-2017 motion), showing strong results.

**Strengths:**

- (S1) The paper includes qualitative examples of the model's performance in a video format, which strengthens the claims and showcases performance.
 - (S2) The quantitative results are strong, and the model is compared to a number of baselines on several benchmarks, which are well discussed.
 - (S3) The ablations, though not extensive, confirm the inclusion of the main loss components.
 - (S4) The proposed combination of B-splines and a 12-parameter quadratic motion model appears novel and interesting.

**Weaknesses:**

The central weakness of the paper is the need for more critical details and more argumentation for the approximations made, particularly in the methods section. The methods section focuses a lot on the ELBO aspect of the argumentation. How B-splines are used to model motion parameters and how they feature in the implementation are only described in passing prose, which is not sufficient given that it is the "key" idea of the paper. In detail:

 - (W1) The paper needs to clearly describe the procedure for B-Spline estimation that is required to perform the update of the loss. Concretely, the argmin of eq. 9 is not described. Moreover, more details are required to state and explain how B-splines are used to model the 12 parameters of the motion model. This is important to both understanding the paper and being able to reproduce the main proposal in the paper.
 - (W2) More details about equation 5 are required. Why is this a reasonable statement? It seems that optical flow is modelled as a set of independent Laplace variables. This is not discussed and might require some motivation. It is also not clear what the "assumption above" refers to.
 - (W3) Similarly, it would also be beneficial to provide more details on why loss in eq. 7 is a reasonable approximation of KL terms of ELBO. This is not discussed.
 - (W4) Equations 6, 8, and 9 might have the wrong sign. The loss is minimized (Eq. 9); however, the loss in eq. 6 is negative because the sum includes terms that are all positive, and the sign is then flipped. This would then increase the l1 error between parametric estimates and input flow, which seems to be the opposite of what is desired.
 - (W5) Given that both terms of ELBO (eq 4.) are approximated, and their signs seem to have flipped (as written in the paper), it is not clear if the resulting loss function of eq. 8 is still a lower bound.

**Questions:**

- Ye et al. [A] similarly use B-splines for modelling motion and deformations and similarly evaluate segmentation tasks on, e.g. DAVIS. It would be interesting to discuss more the differences in the approaches and provide some comparisons.

- Ponimatkin et al. [B] present a method that reasons about full video as well as jointly considering the whole sequence. It would be good to discuss the differences in the approaches.

- The paper claims the method is fast at inference time. Does this include the estimation of optical flow?

As the results are good and the idea _seems_ interesting and novel, I am willing to improve my rating following the rebuttal, with more details about the method (W1-W4) provided.

Additionally, the reference format does not follow the template/guidelines,  which should be addressed (this was ignored for the purposes of providing the review).

---
### References
 - [A] "Deformable Sprites for Unsupervised Video Decomposition", Ye et al. CVPR 2022
 - [B] "A Simple and Powerful Global Optimization for Unsupervised Video Object Segmentation" Ponimatkin et al. WACV 2023

---

> ### Author Response · Authors · 2023-11-17
> **Response to weaknesses and questions**
>
> W1.
> First, as we described in Section 3.1 of the paper, B-Splines are modelling the evolution of the parameters of the motion model over time. In practice, it means that the resulting motion model for a given time $t$ is a weighted average of the close control points (each control point is a quadratic model as given in eq.1 of the paper). The degree of the splines and the frequency of control points along the temporal axis dictate the weights of this average. Figure 2 of the paper illustrates the evolution of motion parameters over time.
>
> As written in eq.2 of the paper, the parametric models induced over time by the splines are a linear combinations of quadratic parametric models. Thus, we can optimize eq.6 to obtain the argmin in eq. 9, which is a convex problem. In practice, we use a quasi-Newton method, more specifically Limited-memory BFGS, as mentionned in Section A.3 - Optimization Implementation - of the Appendix (now Section A.4 in the updated Appendix).
>
> We forgot to mention in the submitted paper that the code will be made available on GitHub upon paper acceptation, which will allow reproducibility of the results. In addition, we have included a repeatability study in Section A.7 of the updated Appendix.
>
> W2. "Assumption above" refers to the assumption that the flow field can be represented by segment-wise parametric motion models. This assumption is introduced at the very beginning of Section 3.2 of the paper. It is the reconstruction fit between the flow and the segment-wise parametric motion model that is defined by a $L_1$ penalty function. Please see Q1 in our response to reviewer 1 for further details.
>
>
> W3. We have written a detailed description of the mathematical development motivating eq.7. Please refer to the new Section A.1 of the updated Appendix in the supplementary file.
>
> W4. Yes, you are right about the wrong sign. This was a typo. We noticed it just after the submission deadline, and it was too late to fix it.
>
> W5. Yes, it is a lower bound, since the sign was unfortunately wrong (see W4). Again, the detailed description of the mathematical development mentioned above (Section A.1 of the updated Appendix in the supplementary file) clarifies this point.
>
> Q1. We will add this reference. There are strong differences between this work and ours. They use B-splines to model the deformation of the image stripes over time. We use B-splines to model the temporal component of the dense flow. Moreover, their model is optimized at test time on each individual video, which is very time consuming, while we optimize our motion model only at training time. We do not need any motion model at test time.
>
> Q2. We will add this reference. In this work, optimization (spectral clustering) is required at test time over the video sequence. In addition, it leverages pretrained appearance features (DINO), which necessarily leads to higher performance in the VOS benchmark. Their goal is actually to achieve video object segmentation, whereas we have developed a motion segmentation method, not a VOS one as such. We use VOS benchmarks for evaluation in the absence of benchmarks for optical flow segmentation. By the way, this also puts us at a disadvantage, because moving objects in the background or motion parallax supplied by static objects in the foreground, both constituting additional motion segments, are not included in the VOS ground truth.
>
> Q3. The computation time at test time of our motion segmentation method is very low (210 fps on average). It does not include the computation of the optical flow, since it is just the input of our method. The computation time for the optical flow estimation depends of course on the optical flow method used. Recent methods are relatively fast.

---

> ### Comment · Reviewer_po8i · 2023-11-21
>
> I want to thank the authors for their response and the updated paper and for taking the time to answer the questions.
>
> > We can optimize eq.6 to obtain the margin in eq. 9, which is a convex problem. In practice, we use a quasi-Newton method, more specifically, Limited-memory BFGS.
>
> Thank you for the explanation. It does clarify and complete the overall picture of how the loss is calculated, including that there is a different convex optimization process. I would like to encourage including this clarification in the methods section if possible.
>
> On the same point:
> are the $\tilde{f}_\theta$ of eq (1) and (2) $\tilde{f}\_{\theta\_{k,l}}$ the same? Eq. (1) seems to take a point $(x,y)$ while (2) seems to take point and time $(i,t)$.
>
> Also, I wanted to check about the mask merging for DAVIS/FBMS/SegTrack evaluation. As the problem seems to be binary segmentation, how are the K=4 masks handled? The appendix seems to indicate that the method follows ST-MS, which seems to perform segment-linking based on IoU with the ground truth masks. Is this correct?
>
> Overall, most of my questions have been answered. I still have a slight minor reservation regarding the clarity of the method section and would also like to consider other Reviewers' discussion.

---

> > ### Author Response · Authors · 2023-11-22
> > **Response to additional comments**
> >
> > We thank the reviewer for these additional comments.
> >
> > C1. We will include this clarification in the final version of the paper.
> >
> > C2. Notation $i$ here represents a point in the 2D image space, and thus, we can extend it as $(x,y)$. However, we should not have included $t$ in the arguments of function $\tilde{f}\_{\theta\_{k,l}}$ in the right member of eq.2, since the temporal dimension is controlled by the spline function $B$ and index $l$. Indeed, the function $\tilde{f}\_{\theta\_{k,l}}$ is defined in the spatial dimensions $x$ and $y$, as in eq.1, where we provide the general expression of the 12-parameter quadratic motion model (therefore, we use in eq.1 the general notation $\tilde{f}\_{\theta}$). We will remove the wrong $t$ in the final version of the paper. We will also replace $i$ by the $x,y$ coordinates.
> >
> > C3. Yes, we performed exactly the same segment selection as in ST-MS (explained now in Section A.5 of the updated Appendix), when we take four masks ($K=4$) whereas the VOS benchmark evaluation is only binary. In ST-MS, the segment selection is based on the IoU with the ground-truth masks. It is a way to deal with the fact that objects undergoing articulated motion are necessarily over-segmented for ST-MS and our method LT-MS, since ST-MS method and ours are performing motion segmentation and not object segmentation. Let us note that this selection is made only once at the sequence level, not frame by frame, to evaluate the temporal consistency of the labeling. For Davis2017 and FBMS59 in the multi-mask evaluation setting, we follow the official protocol instead.
> >
> > C4. We will clarify the points addressed in the reviews in the Method section of the final paper.

---

### Official Review · Reviewer_S5Mv · 2023-10-31

**Soundness:** 3 good
**Presentation:** 2 fair
**Contribution:** 2 fair
**Rating:** 5
**Confidence:** 5

**Summary:**

The model tackles the problem of motion segmentation, which performs pixel-level segmentation on video frames by motion. The motion is first estimated using optical flow, then a stack of optical flow frames is used as input for the model, and the model outputs multiple segmentation masks. To achieve this, the proposed method fits optical flow within each segment using a 12-parameter quadratic motion model and utilizes B-spline to handle the motion's temporal changes. A specific loss function is designed for this purpose, taking into account temporal consistency. This method is trained on synthetic data (FlyingThings3D) and doesn't require ground truth annotation. The results of the model's performance are reported on three standard motion segmentation datasets.

**Strengths:**

The proposed method is fully unsupervised, which is a big advantage in motion segmentation.

The model is able to output multiple region segmentation, in contrast to the majority of motion segmentation methods that only output the foreground segmentation.

Interpreting the motion segmentation loss as Evidence Lower Bound problem is novel and interesting.

The proposed method is lightweight and fast, which is an advantage over other temporal processing methods.

**Weaknesses:**

1. ‘Motion segmentation methods often proceed frame by frame’ is not entirely accurate. Earlier methods, especially before the era of deep learning, were designed to process entire videos. This is supported by a rich body of literature. Additionally, recent methods, such as "Deformable Sprites for Unsupervised Video Decomposition" (CVPR 2022), also operate on the entire video.

2. The paper mentions that it's the "first fully unsupervised" method to incorporate long-term temporal consistency, which is not true. Many non-single image methods, especially those based on layers, e.g. "Layered segmentation and optical flow estimation over time" (CVPR 2012), inherently have long-term temporal consistency built in.

3. The paper suggests that relying solely on optical flow is an advantage. However, optical flow is estimated on image pairs, so the whole process always has images as input. Image intensity provides crucial information for object segmentation, so not being able to use images is actually a drawback, not an advantage.

4. The statement claiming, "We have not only… but also achieved some kind of tracking," is vague. The FBMS-59 dataset is explicitly designed for segmenting and tracking each instance. If the proposed method operates on video sequences and performs temporal association, it would be important to be evaluated under the original settings.

5. The paper predominantly compares with single-frame methods (e.g., EM, MoSeg, DivA, CIS) that do not consider temporal consistency. So the proposed method is expected to yield higher results when using an entire video batch. An exception is the comparison with OCLR. OCLR performs better while the paper argues that OCLR is not fully unsupervised since it trains on synthetic data with labels. Still, the proposed method also trains on synthetic data, where labels are typically sufficient and readily available, and not using the labels is no longer an advantage. The performance of the proposed method is notably worse than OCLR, especially considering that it only compares with the 'flow-only' version of OCLR. If the paper aims to assert the advantage of not using labels, it should directly train on generic real data. See (2.) in Questions.

6. The proposed method is essentially an extension of ST-MS, claiming to introduce temporal consistency over the entire video sequence. However, if I understand it right, the temporal consistency in equation (7) remains constrained to neighboring consecutive frames, similar to ST-MS. The primary distinction lies in using B-spline to model long-term motion changes and a different network architecture.

7. The incorporation of transformers is only valuable if it leads to significant improvements, which do not appear to be impressive (particularly on FBMS-59).

8.  The paper does not discuss any limitations or potential failure modes of the proposed method.

9. Details such as training speed, hardware requirements for training, training batch size, and the size of the training dataset are not provided in the paper. These details are essential information for reproduction.

**Questions:**

1. How is K decided? Is it possible to show results with different K's?

2. Why does the method train on flyingthings3D if it is unsupervised? What does the result look like if trained on DAVIS, Segtrack-V2 and FBMS-59, or the union of them?

3. In Table 1, using more frames leads to worse results, which is counterintuitive.

4. In Figure 3, the motorcycle and rider fly together and should share the same motion pattern. Why are they segmented into different regions?

---

> ### Author Response · Authors · 2023-11-17
> **Reponse to weaknesses and questions**
>
> W1. This statement was not intended to be categorical (use of “often”), and we implicitly referred to the deep learning era (we explicitly did it in the related work section). We are aware of previous classical methods processing the sequence, e.g., by taking the sequence as a space-time volume and providing “tubes”. We will make this point clearer in the abstract and introduction. We will add in Section 2 the mentioned reference (please see Q1 in the response to reviewer 3).
>
> W2. Again, we were referring to the deep-learning era in this sentence. The mentioned reference will be added at the beginning of Section 2.
>
> W3. Our purpose is the segmentation of optical flow fields in long video sequences, not VOS as such. Providing motion segmentation per se is very useful. Our model could be used solely or in cooperation with other modules, e.g., appearance, depending on the downstream task. Regarding VOS, it is not an advantage to use it solely.
>
> W4. Our method processes the entire video in one go with one unique segment labeling over the sequence (this is what we meant by “kind of tracking”). It is a more demanding goal, but when achieved, offers more possibilities. We also evaluated our method under this sequence setting for FBMS59 as reported in Table 3, column “Linear Assignment”, by using the official evaluation method of DAVIS2017 for both datasets.
>
> W5. The use of manual annotation in OCLR to build the training set actually provides a competitive advantage, as previously outlined by the authors of DivA, since the annotation concerns the segmentation itself. On our side, the purpose of using the synthetic FT3D dataset is completely different.  We do not leverage any annotation or segmentation ground-truth. Our (demanding) goal by using FT3D was to demonstrate the high generalization power of our model. We train it once for all on the optical flow fields computed on FT3D, and then, apply it on new unseen datasets. We will add comments on that point to make it clearer.
>
> Motion segmentation at sequence level does not necessarily guarantee a better score on benchmarks where evaluation is done at frame level and does not enhance temporal consistency. In fact, it is more demanding, while at the same time representing a highly beneficial added value for downstream tasks.
>
> We took OCLR “flow-only” for a fair comparison, since we have developed a motion segmentation method, not a VOS one as such. We use VOS benchmarks for evaluation in the absence of benchmarks for optical flow segmentation.
>
> W6. The differences with ST-MS are significant. First, they concern the core ingredients: motion model and network architecture. Second, the introduction of the B-spline temporal motion model is quite novel and decisive. Thanks to this motion model and the use of transformers, we were able to design an end-to-end learning-based method, in contrast to ST-MS that requires a complex postprocessing for temporal linkage.
>
> W7. Transformers helped in defining an end-to-end method, which is essential beyond improving performance. Adding the transformer to the latent space of the Unet allows for interaction between distant features in the input, and thus a better long-term segmentation. Furthermore, we only incorporated a transformer decoder on the downsampled latent space of the Unet, which induces a limited additional computational load.
>
> W8. Failure cases may occur for very complex motion like fluid motions (e.g., ripples on the water) or with a particularly strong temporal evolution (e.g, an object approaching the camera from afar quickly and closely). Furthermore, as shown in Fig.3 of the paper and Fig.6 in the Appendix, our method is robust to errors in the flow field, provided they occur locally in time (over a few frames), but could fail in case of a long-term perturbation. We will add this comment.
>
> W9. Requested details on the training stage will be added in Section 4.1. Details on the size of each dataset along with the split into training and test were given in Section A.1 of the Appendix (now Section A.2 of the updated Appendix). We forgot to mention in the submitted paper that the code will be made available on GitHub upon paper acceptation, allowing reproducibility. In addition, we have included a repeatability study in Section A.7 of the updated Appendix.
>
> Q1. K is set by the user before training. In fact, results with different K values are provided (K=2 and K=4 in Table 2, K=3 in Table 3).
>
> Q2. Please see W5. In addition, we report training on DAVIS2016 train set in Section A.8 of the updated Appendix, and results are on par.
>
> Q3. At first glance, it could be counterintuitive, but it is due to the way the evaluation is performed in these benchmarks (see W5).
>
> Q4. Yes, the motorcycle and rider fly together, but they undergo slightly different motions as perceived in the flow fields. The motorcycle wheel turns a little. Furthermore, our method operates on the vector field, not on its HSV color representation.

---

### Official Review · Reviewer_Wuh7 · 2023-11-02

**Soundness:** 2 fair
**Presentation:** 2 fair
**Contribution:** 2 fair
**Rating:** 5
**Confidence:** 3

**Summary:**

The paper presents an approach for a long-term motion segmentation problem in videos. The work assumes a readily availably optical flow as an input data. Subsequently, a parameterized motion model that seems to decompose optical flow as linear combination of several estimates and a temporal consistency term is used for motion segmentation.

**Strengths:**

- Although I am not sure I fully understand the motion model, decomposing the optical flow in some rate of change space seems to be a good idea. I suggest the authors to look into the ideas of "Slow feature Analysis"
- Experimental results show improvement due to the approach.

**Weaknesses:**

The problem formulation and the suggested solution is a little bit difficult to understand. Hence, I struggled to identify its strength/weakness. Please see questions for more details. I recommend re-writing of the Section 3.

**Questions:**

**Motion model**.
- What exactly is the motion model. It seems to be some kind of linear decomposition of the optical flow. is that correct? If so please clarify and denote all notation in Eq(2), for instance what is $n$?
- Can you include label for the axis in Figure 2? I assume, the x-axis, represent frame index while y-axis is are the linear coefficients

**The loss function and general model**.
-  The loss function is a little bit difficult to understand. As far as my understanding goes it is unsupervised model that attempts to estimate the data model $p(x)$ via ELBO. A couple of questions here
      1. What is the model used to estimate the likelihood $p(x|z)$ in eq(5), it is not clear what kind of parametric model or the kind of estimator used?
      2. what is the form of the priori $p(z, \theta)$ ? it is not quite clear. In eq(7), a regularize that penalizes difference in subsequent motion model predictions is added, but is not clear how this relates to $KL$ between posterior and prior.

- The above questions are particularly important, because the framing of the problem as ELBO estimation leads to a generative model with recognition capacity. In such a case, we can discuss further on how capable either the recognition part (posterior estimation) or reconstruction part are depending on the application. I am not sure if the work presented such a model. In my opinion, the work is best described as optical-flow based motion estimation model with regularization.

---

> ### Author Response · Authors · 2023-11-17
> **Response to questions**
>
> Motion model
>
> Q1. The parametric motion model is a polynomial expansion in the coordinates x and y of any point in the image. There is one polynomial for each component of the 2D flow vector. Polynomial expansion is a usual approximation of any unknown function.  It is linear w.r.t. to the polynomial coefficients. However, it is not a linear decomposition of the optical flow for polynomial of degree greater than 2. In eq.2, $n$ is the degree of B-spline (we take $n=3$ as specified at the beginning of Section 3.1 of the paper), $t$ denotes time. The other notations are already defined just above or just below eq.2.
>
> Q2. The $x$-axis in Fig.2 is the frame index in the sequence and the $y$-axis the value of each parameter of the motion model (six parameters for the polynomial of degree 2 representing the horizontal motion component). This will be added in the figure caption.
>
>
> Loss function.
>
> Q1 : Likelihood $p(f|z,\vartheta)$ is a reconstruction model where the observed input flow $f$ within each segment $z_k$ is represented by a parametric motion model with parameters $\vartheta_k$. We assume conditional independence of the flow vectors (which is a usual assumption in motion analysis) to derive the final expression in eq.5. In addition, we use a robust penalty function to discard possible outliers, that is, the $L_1$ norm, as a similarity measure.
>
> Q2. We have written a detailed description of the mathematical development that motivates eq.7 including the expression of the prior $p(z,\theta)$. Please refer to the new Section A.1 of the updated Appendix in the supplementary file.

---

### Meta-Review · Area_Chair_V9Qj · 2023-12-04

**Metareview:**

The paper presents an approach for a long-term motion segmentation problem in videos. The work assumes a readily availably optical flow as an input data. Subsequently, a parameterized motion model that seems to decompose optical flow as linear combination of several estimates and a temporal consistency term is used for motion segmentation.

Pro
* novelty of the method,
* The rebuttal and revision addressed several concerns in the initial reviews: such as "absence of implementation details and hyperparameters, insufficient discussions on limitations and failure modes, and the use of synthetic data in training without ample justification instead of real data.", and "presentation issues"

Con/However, there are remaining concerns:
* Motivation. "Positioning the paper towards the task of "Optical Flow Segmentation" (OFS) instead of generic motion segmentation raises questions." the paper needs re-positioning about the contribution "the work is not the first to address motion segmentation in one go."
* Performance "performance of the proposed method does not significantly surpass baseline methods. Most baselines in the paper operate on single images and flow, whereas the proposed methods process multiple frames, providing additional segmentation information."
* the claim "in one go," versus batching images into a group of 10.

**Justification For Why Not Higher Score:**

There are remaining concerns after the rebuttal and discussion regarding motivation/positioning, performance and claim of the paper.

**Justification For Why Not Lower Score:**

N/A

---

### Decision · Program_Chairs · 2024-01-16

Reject